# High Throughput Sequencing-Aided Survey Reveals Widespread Mixed Infections of Whitefly-Transmitted Viruses in Cucurbits in Georgia, USA

**DOI:** 10.3390/v13060988

**Published:** 2021-05-26

**Authors:** Saritha Raman Kavalappara, Hayley Milner, Naga Charan Konakalla, Kaelyn Morgan, Alton N. Sparks, Cecilia McGregor, Albert K. Culbreath, William M. Wintermantel, Sudeep Bag

**Affiliations:** 1Department of Plant Pathology, University of Georgia, Tifton, GA 31793, USA; SarithaRama.Kavalappara@uga.edu (S.R.K.); Hayley.Milner@uga.edu (H.M.); nagacharan.konakalla@slu.se (N.C.K.); Kaelyn.Morgan@uga.edu (K.M.); spotwilt@uga.edu (A.K.C.); 2Department of Entomology, University of Georgia, Tifton, GA 31793, USA; asparks@uga.edu; 3Department of Horticulture, University of Georgia, Athens, GA 30602, USA; cmcgre1@uga.edu; 4United States Department of Agriculture-Agricultural Research Service, Salinas, CA 93905, USA

**Keywords:** cucurbit chlorotic yellows virus, cucurbit leaf crumple virus, cucurbit yellow stunting disorder virus, high throughput sequencing, whitefly transmitted viruses, survey, cucurbits

## Abstract

Viruses transmitted by the sweet potato whitefly (*Bemisia tabaci*) have been detrimental to the sustainable production of cucurbits in the southeastern USA. Surveys were conducted in the fall of 2019 and 2020 in Georgia, a major cucurbit-producing state of the USA, to identify the viruses infecting cucurbits and their distribution. Symptomatic samples were collected and small RNA libraries were prepared and sequenced from three cantaloupes, four cucumbers, and two yellow squash samples. An analysis of the sequences revealed the presence of the criniviruses cucurbit chlorotic yellows virus (CCYV), cucurbit yellow stunting disorder virus (CYSDV), and the begomovirus cucurbit leaf crumple virus (CuLCrV). CuLCrV was detected in 76%, CCYV in 60%, and CYSDV in 43% of the total samples (*n* = 820) tested. The level of mixed infections was high in all the cucurbits, with most plants tested being infected with at least two of these viruses. Near-complete genome sequences of two criniviruses, CCYV and CYSDV, were assembled from the small RNA sequences. An analysis of the coding regions showed low genetic variability among isolates from different hosts. In phylogenetic analysis, the CCYV isolates from Georgia clustered with Asian isolates, while CYSDV isolates clustered with European and USA isolates. This work enhances our understanding of the distribution of viruses on cucurbits in South Georgia and will be useful to develop strategies for managing the complex of whitefly-transmitted viruses in the region.

## 1. Introduction

Vegetables are the fourth largest commodity group grown in Georgia by value, and cucurbits (*Cucurbitaceae* Juss.) contribute a total of USD 322 million towards the Georgia farm gate value [1]. Commercial production of cucurbits, including cantaloupe (*Cucumis melo var. cantalupensis* Naudin), cucumber (*Cucumis sativus* L.), and yellow squash and zucchini (*Cucurbita pepo* L.), is concentrated in southern Georgia. In recent years, cucurbit production has become very challenging due to unusually high populations of whiteflies (*Bemicia tabaci* complex), and an associated heavy incidence of whitefly-transmitted viruses, primarily in the fall. Based on crop loss estimates in 2016 and 2017, several millions of dollars were lost due to whiteflies and the viruses they transmit in cucurbits (cantaloupe, yellow squash, and zucchini) and beans (*Phaseolus vulgaris* L.) [2]. Whitefly-transmitted viruses reported from the state, and infecting cucurbits include cucurbit leaf crumple virus (CuLCrV) [3], cucurbit yellow stunting disorder virus (CYSDV) [4], sida golden mosaic virus (SiGMV) [5], and cucurbit chlorotic yellows virus (CCYV) [6]. Recently, the thrips-transmitted tobacco streak virus was also identified on yellow squash in Georgia [7]. The prevalence and distribution of these viruses and their individual contribution to the recent virus outbreaks are not known. Moreover, since these viruses were identified in the state using methods specific to detect each of them, the presence of unsuspected or novel viruses in the region is also not known. In addition to these viruses, the squash vein yellowing virus (SqVYV; genus *Ipomovirus*) is considered a potential threat to cucurbits and melons [8] in Georgia due to its presence in the neighboring state of Florida, where heavy incidence of this virus has been reported.

An analysis of viral populations in plant samples with high throughput sequencing (HTS) has become an established method for detecting and identifying plant viruses and viroids introduced to an agroecosystem [9,10,11,12,13,14,15,16,17]. These methods can potentially detect all viruses and viroids present in a sample, including those previously unknown. HTS has also been used to combine diagnostics with sequence analyses of viruses present in the samples. Different types of nucleic acids (total nucleic acid, total RNA, dsRNA, and siRNA) are used for HTS to enrich viral sequences and minimize interference due to host sequences. Among them, siRNA sequencing has the clear advantage of detecting all types of viral and viroid genomes [18,19]. This approach exploits the natural anti-viral defense system called RNA silencing or RNA interference (RNAi) present in all eukaryotic organisms and generates 21–24 nt small RNA (sRNA) corresponding to the invading viruses [20]. Virus-specific siRNAs are abundant and can represent up to 30% of total small RNAs sequenced from diseased plants [21]. These can be sequenced, isolated, and assembled to identify viruses by homology to known viruses [22,23,24].

In this study, surveys were conducted in the fall of 2019 and 2020 in the cucurbit growing region of South Georgia, and viruses present in these samples were identified by HTS. Symptomatic samples were collected, and nine small RNA libraries were constructed and sequenced from cantaloupe, cucumber, and yellow squash. The prevalence and distribution of the viruses identified in the region were determined by the HTS analysis of a larger number of samples of the three crops collected from different counties in Georgia and employing conventional detection methods. Near complete genome sequences of viruses were assembled from HTS data, and their characteristics and phylogeny were determined. Using the unbiased small RNA sequencing and de-novo assembly, we identified the whitefly-transmitted viruses infecting cucurbits in the region and showed that they are more widely distributed than previously assumed.

## 2. Materials and Methods

### 2.1. Survey and Sample Collection

Colquitt, Tift, and Worth counties, together represent the major cucurbit-producing areas of Georgia. A survey of the main fall cucurbit crops in the state (cantaloupe, cucumber, yellow squash, and zucchini) was conducted in commercial fields in Colquitt and Worth counties and at the experimental farms of UGA, Tifton in Tift County in the fall of 2019 and 2020 (Appendix A). A minimum of 20 cucurbit leaf samples showing virus-like symptoms were collected from each field and kept on ice until processing. A subset of symptomatic leaf tissues was washed with distilled water within the field and frozen immediately on dry ice for use in the small RNA analysis. All samples were brought to the Plant Virology Laboratory at UGA Tifton and stored at −80 °C until further analysis.

### 2.2. Small RNA Sequencing from the Samples

Samples frozen on dry ice in the field were shipped to Beijing Genomics Institute (BGI, San Jose, CA, USA) for sequencing. Small RNA libraries were constructed from two samples each of cantaloupe, cucumber, and yellow squash. The libraries were sequenced on a DNA Nanoball (DNB) small RNA sequencing platform, single-end read 1 × 50 bp (BGI, Hong Kong).

### 2.3. Detection of Viruses in HTS Data

Small RNA processing, assembly, virus detection and identification in the samples were carried out using CLC Genomics Workbench 21 (Qiagen, Redwood City, CA, USA). Adapter sequences, low-quality sequences, and sequences with more than two ambiguous nucleotides were trimmed. The reads were filtered on their size to retain reads of length between 18–30 nucleotides. To enrich virus-derived sequences, host sequences were removed by mapping from the cleaned and trimmed reads of the host genome (Cantaloupe- Melon_v4.0_PacBio; cucumber-Gy14_cucumbergenome_v2; squash-Cpepp_genome_v4.1). The host genomes were retrieved from the Cucurbits Genomics Database (CuGenDB) [25]. Contigs were assembled de-novo from the host subtracted reads following the parameters outlined by Pecman et al. (2017) [26]. Contigs that were less than 50 nucleotides in length were not included in the analysis. A local viral database was created (http://www.ncbi.nlm.nih.gov/genome/viruses) (downloaded on 14 December 2020) from the National Center for Biotechnology Information (NCBI) using the Create Database feature of the CLC Genomics Workbench 21. These contigs were compared for similarity using BLASTn [27] against all sequences in the database with default parameters set in the CLC Genomics Workbench 21. Contigs mapping to bacterial and fungal genomic fragments or non-plant viral sequences were not analyzed further.

### 2.4. Detection of Viruses by Conventional Methods

Total nucleic acid (TNA) was isolated from symptomatic samples using magnetic bead technology to ensure high-quality nucleic acids that could be used in both PCR and RT-PCR to detect DNA and RNA viruses, respectively. Samples were homogenized in a 4 M Guanidine thiocyanate (GTC) buffer (pH 5.0) by mechanical disruption. Thirty milligrams of symptomatic plant tissue were added in 2 mL screw-cap tubes with 200 µL of GTC buffer and two ceramic beads 2.8 mm each (Omni International, Kennesaw, GA, USA). Tissue was homogenized using a Bead Mill 24 Homogenizer (Thermo Fisher Scientific, Waltham, MA, USA) at a speed of 3.55 m/s for 45 s. The homogenate was centrifuged (10,000× *g* for 30 s), and the supernatant was collected. The supernatant (115 µL) was mixed with magnetic beads and processed with the MagMAX 96 viral RNA kit using the KingFisher Flex Purification System (Thermo Fisher Scientific, Waltham, MA, USA) following the manufacturer’s instructions excluding the DNase treatment. TNA was eluted with 100 µL nuclease-free water. The quality and quantity of nucleic acids were determined using a NanoDrop One Microvolume UV-Vis Spectrophotometer (Thermo Fisher Scientific, Waltham, MA, USA). TNA with 230/280 absorbance of >1.8 was aliquoted for further analysis.

The presence of viruses identified from each sample was confirmed using PCR or RT-PCR. The cDNA was prepared with random hexamer primers (Thermo Fisher Scientific, Waltham, MA, USA) followed by PCR with virus-specific reverse primers (Table 1). For cDNA preparation, 10 µM random primer-0.5 µL was mixed with 5 µL TNA and denatured at 70 °C for 5 min and immediately chilled on ice. The reverse transcription master mix contained 2 µL of RT buffer, 0.5 µL of 100 mM DTT, 1 µL of 10 mM dNTP mix, 0.5 µL Superscript III (200 U/µL) (Invitrogen, Carlsbad, CA, USA), 0.25 µL RNase inhibitor (50 U/µL), and RNAse free water to a total volume of 10 µL. The reaction mix was incubated at 42 °C for 1 h, followed by 25 °C/5 min and 37 °C/20 min.

The polymerase chain reaction (PCR) was carried out with 5 µL of 5 X GoTaq green buffer (w/MgCl_2_), 0.5 µL dNTPs (10 mM), 0.5 µL reverse primer (10 µM), 0.5 µL forward primer (10 µM), 15.75 µL RNAse free water, 0.25 µL GoTaq polymerase (5 U/µL) (Promega, Madison, WI, USA), and 2.5 µL cDNA or TNA to a final volume of 25 µL. Cycling conditions for all primers were initial denaturation at 95 °C for 2 min, followed by 35 cycles each of 95 °C for 30 s, annealing temperature depending on primers for 30 s (Table 1), 72 °C for 1 min/kb, and final extension of 72 °C for 5 min in the T100 thermal cycler (Bio-Rad, Hercules, CA, USA). Plasmids carrying the fragments of the virus amplified by the primers used for testing and no template (water) were used as positive and negative controls, respectively. PCR products were analyzed on 1% agarose gel in 1X TAE containing Gel Red (Biotium, Fremont, CA, USA).

### 2.5. Detection of DNA Viruses by Rolling Circle Amplification and HTS

The rolling circle amplification (RCA) was carried out using random hexamer primers to amplify the circular DNA viruses present in the samples [29]. RCA products from one cantaloupe, two cucumbers, and three squash samples collected in 2019 were sequenced (2 × 150 bp paired ends) on an Illumina HiSeq 1500 system at the Georgia Genomics and Bioinformatics Core (GGBC), Athens, UGA. To identify begomoviruses from RCA products, the protocol used for detection of RNA viruses was followed with the exception that the size of contigs included for analysis was increased to 250 bp.

### 2.6. Construction of Consensus Viral Genome Sequence and Coverage Maps

Consensus sequences were assembled by reference-based mapping using CLC Genomics Workbench 21. Reference sequences of viruses potentially present in the samples were downloaded from NCBI (CuLCrV: DNA A-NC_002984, DNA B-NC_002985; CCYV: RNA1-NC_018173.1, RNA2-NC_018174.1; CYSDV: RNA1-NC_004809.1, RNA2-NC_004810.1). Small RNA reads were aligned with the reference sequence. Consensus sequences were assembled from the mappings with the following parameters: Mismatch cost = 2, insertion cost = 3, and deletion cost = 3. For graphical visualization of coverage in each region of the assembled genome, read tracks showing maximum, minimum, and average coverage values were created. The alignment of consensus sequences with the reference genome was inspected for discrepancies with CLC Genomics Workbench 21 and Bioedit [30]. The NCBI-ORF Finder (https://www.ncbi.nlm.nih.gov/orffinder/, accessed on 2 February 2021) was used to identify ORFs in the assembled sequences and to translate proteins from the consensus sequences.

### 2.7. Phylogenetic Analysis

Consensus sequences of CCYV and CYSDV were compared with reference sequences of all the criniviruses downloaded from NCBI GenBank. Multiple sequence alignments were performed using MUSCLE [31] in the Molecular Evolutionary Genetics Analysis, version X (MEGA X) [32]. The evolutionary history was inferred using the neighbor-joining method [33]. Bootstrap values were calculated using 2000 random replications and condensed to 50%. The calculated trees were displayed using Tree Explorer implemented in the MEGA X program. The percentage differences between the nucleotide sequences of the isolates were determined by a multiple sequence alignment using MUSCLE in the Sequence Demarcation Tool software, version 1.2 [34].

## 3. Results

### 3.1. Symptomatology

Virus-like symptoms were observed in all the cucurbit farms surveyed in Colquitt, Tift, and Worth counties. Yellow squash was the most severely affected crop based on the severity of symptoms as well as the disease incidence observed in the field. The different types of symptoms observed on yellow squash plants included chlorosis, crinkling, vein yellowing, interveinal leaf chlorosis, yellow spots, and severe stunting or bunching of leaves at the top of the plant (Figure 1A–E). Fruits, if any were formed on those yellow squash plants, displayed severe bunching, were distorted, and were streaked with green patches (Figure 1F). In every farm visited in 2020, all plants showed at least one of these symptoms with varying degrees of severity.

In contrast, less severe symptoms and lower disease incidences (10–30%) were observed in commercial farms of cucumber, cantaloupe, and zucchini. The main symptom observed on cantaloupe and cucumber was interveinal leaf chlorosis, more prominent at the crown region (Figure 1G–I) and milder on the younger leaves (Figure 1J). Leaf crinkling and stunting symptoms displayed on yellow squash were not observed on cantaloupe and cucumber. Foliage symptoms on zucchini including crinkled leaves and a yellow mosaic pattern (Figure 1K–L), was observed in commercial cucurbit farms. The viruses that were identified in this study and mentioned in the following sections exhibit overlapping symptoms and are difficult to differentiate in mixed infection under natural conditions.

### 3.2. Virus Detection by Metagenomics

Nine small RNA libraries were prepared and sequenced from three cantaloupes, four cucumbers, and two yellow squash samples representing the dominant symptoms observed in the fields. Between 12 and 22 million reads were retained in each of the samples (Appendix A) after trimming and quality control. A similar length distribution of small RNA was observed in all the libraries. The most abundant class of small RNAs were 21–24 nt in length (Appendix A). Data from each sample was analyzed separately. Sequences that did not map to the host genome were assembled into contigs with a minimum cut off size of 75 bp and compared with virus sequences in NCBI using BLASTN for their identification. Sequences of bacterial and fungal viruses identified from the samples along with those of plant viruses, were not analyzed.

Among plant viruses, both DNA and RNA viruses were detected primarily as components in multiple virus infections (Figure 2A). All genomic components of RNA1 and RNA 2 of CCYV and CYSDV, as well as DNA A and DNA B of CuLCrV were detected from cantaloupe, cucumber, and yellow squash. The near-complete genomes of these three viruses were also assembled from all the crops with high coverage for most genomic regions, giving high confidence for their presence (Figure 3). Sequences suggestive of other begomoviruses (Figure 3) were detected. However, their genome coverage was poor, with fewer reads mapped on the reference genome compared to those for CCYV, CYSDV, and CuLCrV (Figure 3). In the analysis of HTS of RCA products from the virus infected cantaloupe, cucumber, and squash, only CuLCrV could be detected thus effectively ruling out the presence of any other DNA viruses (Figure 2B).

The small RNA datasets from all three crops were also examined for cucurbit viruses reported in the USA and causing similar symptoms to those observed in samples subjected to NGS by mapping reads on their reference genomes. These included cucurbit aphid-borne virus (CABYV), squash vein yellowing virus (SqVYV), cucumber vein yellowing virus (CVYV), and zucchini yellow mosaic virus (ZYMV). However, none of these viruses were detected in any of the samples.

### 3.3. Virus Prevalence and Distribution among Different Cucurbits

In 2019 and 2020, symptomatic samples were collected from fall cucurbits in the major vegetable growing region located in the southern part of the state of Georgia. The crops surveyed included cantaloupe, cucumber, yellow squash, and zucchini. Samples were tested by RT-PCR and PCR assays to determine the extent of distribution of the viruses identified by HTS in this study and those previously reported from Georgia (CCYV, CYSDV, CuLCrV, and SiGMV).

Whitefly-transmitted viruses including CCYV, CYSDV, and CuLCrV were detected on all the crops tested and from all counties surveyed. Among a total number of 820 symptomatic samples, CuLCrV was detected on 76%, CCYV on 60%, and CYSDV on 43% of the samples tested (Table 2). The level of mixed infections was very high in all the cucurbits, with most of the samples infected with at least two viruses. Dual infections of CuLCrV and CCYV were detected on 44%, CuLCrV and CYSDV on 33%, CCYV and CYSDV on 33% of the samples tested. Mixed infections of all three viruses, CCYV, CYSDV, and CuLCrV, were detected in 23% of samples (Table 2).

Even though CuLCrV, CCYV, and CYSDV were detected on all cucurbit types tested, there was a difference in their relative prevalence in different crops. On cantaloupe and cucumber, either one of the criniviruses (CCYV or CYSDV) was detected at a higher frequency than the begomovirus (CuLCrV) (Table 2). On the other hand, yellow squash had a higher frequency of CuLCrV followed by CCYV and CYSDV, although the frequency of criniviruses increased in 2020 compared to 2019. Zucchini samples were only collected and tested in 2020. All three viruses were detected at high frequency in zucchini (Appendix A). The Sida golden mosaic virus was not detected on any of the samples tested in 2019 and 2020. All samples collected in 2020 were tested for CABYV and SqVYV but neither of these viruses was detected.

### 3.4. Phylogenetic Relationships of Cucurbit Chlorotic Yellows Virus and Cucurbit Yellow Stunting Disorder Virus Isolates from Georgia

The near-complete genomes of CCYV and CYSDV were assembled from all the nine small RNA datasets. There were gaps in 3′ and 5′ non-coding regions of the assembled sequences for both CCYV and CYSDV genomes and these regions were not analyzed. One sequence of CCYV each from cucumber, cantaloupe, and yellow squash, and one sequence of CYSDV from cantaloupe and cucumber without gaps in the region spanning the coding regions, were used to determine phylogeny of these two viruses from Georgia. Assembled sequences of CYSDV from yellow squash had gaps in the coding regions and were not included in the analysis.

Isolates of CCYV from cantaloupe (RNA 1: 8284 bp, MW629379; RNA 2: 6786 bp, MW629380), cucumber (RNA1: 8284 bp, MW629381; RNA 2: 8041 bp, MW685456), and yellow squash (RNA 1: 8284 bp, MW685455; RNA 2: 6804 bp, MW685461) sequenced in this study were highly similar and shared 99.5% identity at the nucleotide level with each other and the reference sequence of CCYV (RNA1-NC_018173.1, RNA2-NC_018174.1) (Figure 4). The genomic position, size, and sequence homology of all ORFs identified were similar in all three isolates and matched with that of the type isolate. The isolates of CCYV from Georgia were most closely related to the Japanese isolate (RNA1-NC_018173, RNA2-NC_018174.1) in phylogenetic analysis based on the nucleotide sequence of the coding regions of RNA 1 and RNA 2.

Similarly, CYSDV isolates from cantaloupe (RNA 1: 9123 bp, MW685460; RNA 2: 6575 bp, MW685459) and cucumber (RNA 1: 9123 bp, MW685457; RNA 2: 6575 bp, MW685458) in Georgia shared more than 99% identity with one another and with other CYSDV isolates present in the GenBank. There were only three complete sequences of RNA 1 and RNA 2 of CYSDV, one from Arizona and two from Spain. The CYSDV isolates from Georgia were more closely related to the isolates from Arizona and Spain based on the analysis of RNA 1 and RNA 2 coding regions (Figure 4) and diverged from isolates from Saudi Arabia in the analysis based on the partial coat protein gene sequence used for comparison (Appendix A).

## 4. Discussion

In recent years, cucurbit production in Georgia has been severely impacted due to the heavy incidence of whiteflies and the viruses transmitted by them, primarily in the fall. There is no actual estimate of losses in dollars due to the complexity of whitefly-transmitted virus diseases, but the amount is believed to be in the tens of millions of dollars [2]. Two whitefly-transmitted viruses (WTVs), CuLCrV [3] and CYSDV [4], were previously reported to infect cucurbits in the state, whereas CCYV was first reported in Georgia this year (2021) [6].

The bipartite DNA virus, CuLCrV belongs to the genus *Begomovirus* and is persistently transmitted by the *B. tabaci* complex [35]. First discovered in the Imperial Valley of California in 1988 [36], CuLCrV has spread to other parts of the United States. It was reported from Florida in the southeastern USA in 2006 [37] and more recently in South Carolina [38]. In Georgia, it was first identified on snap beans (*P. vulgaris*) in 2010 [3] and had been causing severe damage to yellow squash and snap beans in the state. In yellow squash, disease symptoms are severe and include stunted growth, as well as curled and crumpled young leaves. Fruits of yellow squash develop green streaks [39]. With a wide host range, CuLCrV can infect many plant species belonging to *Cucurbitaceae* and *Fabaceae* [40,41].

First reported in the United Arab Emirates in the early 1990s [42], CYSDV has spread globally to regions in Asia, Europe, North Africa, and North America over the past two decades [43,44]. In the new world, CYSDV was initially reported in southern Texas and northern Mexico [45], the Imperial Valley of California, and Yuma, Arizona [46]. It became widespread in the western parts of the Sonoran Desert in Arizona and Sonora, Mexico [47]. Soon after, it was detected in the southeastern United States in Florida [48]. In Georgia, CYSDV was first identified on yellow squash showing yellowing and green vein symptoms [4]. CYSDV can infect members of the *Cucurbitaceae*, alfalfa (*Medicago sativa*), lettuce (*Lactuca sativa*), certain cultivars of snap beans (*P. vulgaris*), and many weed species [49]. CYSDV infection in squash is characterized by severe interveinal chlorosis, especially in the older leaves, and can result in significant decreases in sugar production in melons, rendering them nonmarketable.

The consistent detection of CuLCrV and CYSDV on symptomatic cucurbit samples led to the assumption that these were the only viruses responsible for losses in the state. However, another crinivirus, CCYV that produces symptoms virtually identical to those of CYSDV, was detected recently on squash grown in Tift County [6]. Discovered in Japan in 2004 [50], CCYV was believed to be restricted to Asia [50,51], Africa [52], and the Mediterranean regions of Europe [53,54] until it was recently identified in the Imperial Valley of California [55]. In addition to the Cucurbitaceae, the experimental host range of CCYV includes species of the *Asteraceae*, *Chenopodiaceae*, *Convolvulaceae*, *Solanaceae*, including weeds and alfalfa [56,57].

CCYV and CYSDV belong to the genus *Crinivirus* and are part of an emerging complex of largely whitefly-transmitted viruses associated with cucurbit yellows disease [49,56] and responsible for worldwide losses of billions of dollars annually [58]. Both these viruses are transmitted by *B. tabaci* MEAM1 and MED [56], although MED is more effective in the transmission of the CCYV than MEAM1 [59].

Losses due to CuLCrV and CYSDV are reported only on squash and beans in Georgia [2]. The extent and spread of these recently identified viruses in commercial fields and among other cucurbits in the region which can serve as reservoir hosts, is unknown. These additional hosts are likely an essential link in the survival and increasing disease incidence caused by these viruses in the region.

Most damage due to WTVs is reported during the fall season, therefore, surveys were conducted during the fall production season. The damage caused by viral diseases was most severe on yellow squash. All plants were apparently infected with at least one virus on all farms surveyed. The most striking symptom observed on squash was the bunching of leaves at the top of plants and severely distorted fruits with green streaks (Figure 1). CuLCrV was detected on 92% and 85% of squash samples tested in 2019 and 2020, respectively (Table 2). Criniviruses were detected in 2019 and 2020 on squash, although the percentage of samples with crinivirus infections increased in 2020. CCYV was detected in 82% of samples, while CYSDV was detected in 50% of the samples in 2020.

Plant viruses have been shown to influence the behavior and physiology of their vectors to increase the efficiency of their transmission. CCYV has been shown to affect the feeding behavior of *B. tabaci*, which may increase the ability of its vector *B. tabaci* to transmit CCYV [60]. In the CCYV and CYSDV pathosystem on cucumber, nonviruliferous whiteflies preferentially settled on virus-infected host plants whereas viruliferous whiteflies were more attracted to healthy plants [61], thus making effective use of an existing source of inoculum for initiating secondary spread. In contrast, in the yellow squash viral pathosystem (CuLCrV and/or CYSDV), both non-viruliferous and viruliferous whiteflies have been shown to preferentially settle on non-infected plants [61]. In the southeastern United States, an overwhelming number of whiteflies disperse from plants that are not hosts of these viruses (such as cotton) to vegetable farmscapes during fall and acquire one or more of the viruses that can infect squash. Preferential settling of these viruliferous whiteflies on non-infected plants can then enhance the virus spread of one or more viruses in the susceptible crops causing heavy incidence of these viruses [62].

On cantaloupe and cucumber, 20–30% incidence of virus infections were found in all the locations surveyed. The main symptom observed was interveinal chlorosis, but the damage was not as severe as those observed on squash. Leaf crumple symptoms were not prominent on cantaloupe or cucumber in any of the commercial fields and farms surveyed. Another interesting observation was that in both years, either of the criniviruses (CCYV or CYSDV) was detected at a higher frequency compared to CuLCrV on cantaloupe and cucumber. On cantaloupe, CCYV was detected in 100% of the samples collected in 2019, while CYSDV was detected in 90% of the samples collected and tested in 2020. CuLCrV was detected in 29% in 2019 and 53% of samples tested in 2020 (Table 2). On squash, in both 2019 and 2020, CuLCrV was detected at higher frequencies than any of the criniviruses although the extent of difference varied. On zucchini plants, leaf crumple and interveinal yellowing symptoms were observed, and CuLCrV, CCYV, and CYSDV were frequently detected as mixed infections. There was also a difference in the distribution of viruses on zucchini at different locations. CuLCrV was detected in a higher percentage of plants than CCYV and CYSDV on zucchini in Worth County.

Virus-virus interactions associated with mixed infections can shape the population dynamics of component viruses [63,64,65] in a host. A few studies have illustrated the crop-dependent preferential accumulation and transmission of one of the viruses among CCYV, CYSDV and CuLCrV over others during mixed infections. CYSDV accumulated in significantly lower amounts in yellow squash plants infected with both CCYV and CYSDV than those infected with only CYSDV. Whiteflies acquired similar levels of CuLCrV but reduced levels of CYSDV from mixed-infected squash plants in comparison to plants infected with only any one of these viruses [62]. This may partially explain the higher numbers of CuLCrV infected plants than any of the criniviruses, which were observed on yellow squash in this survey in 2019 and 2020. In melon (cantaloupe), on the other hand, the rates of whitefly transmission of CYSDV increased when plants dually infected with CYSDV and the potyvirus, watermelon mosaic virus (WMV) were used as a source for virus acquisition [66]. In cucumber, the accumulation of both CCYV and CYSDV and subsequent transmission efficiency of each of these viruses by whiteflies were significantly decreased during mixed infections compared to the results during single infections. However, their simultaneous transmission efficiency was significantly higher [61]. However, their findings cannot completely explain the differences in virus frequencies which were observed on cantaloupe and yellow squash in this study.

Cantaloupe, cucumber, and zucchini are also affected with WTVs, although losses have not been as severe as in squash to date. Detection of criniviruses in fall season crops in 2 consecutive years suggests that these viruses are established in alternative host plants that survive over the winter months.

The recovery of near-complete genome sequences of CCYV and CYSDV isolates from small RNA sequences collected from infected leaves of cucurbit plants grown in Georgia allowed for their molecular characterization and comparison to known sequences. The genomes were assembled with a high degree of confidence since the read coverage was very high for every region of the genome (Figure 3). The near-complete genomes of three isolates of CCYV, one each from cantaloupe, cucumber, squash, and two isolates of CYSDV, one each from cantaloupe and cucumber, were assembled. Since there were a few gaps in 3′ and 5′ non-coding regions, those regions were not analyzed. An analysis of the coding regions of these viruses showed low genetic variability among different isolates of both CCYV and CYSDV, as has been found consistently for these viruses [58,67]. All isolates of CCYV and CYSDV shared more than 98% identity with one another at the nucleotide level among each collected and examined sample. In phylogenetic analysis, the CCYV isolates from Georgia clustered with the Asian isolates of CCYV, while CYSDV clustered with European and USA isolates (Figure 4).

In general, criniviruses show limited genetic diversity even among geographically distant isolates [54,68,69,70]. CYSDV has shown a high degree of genetic stability among all coding and non-coding regions for isolates collected from different cucurbit hosts in Spain [67]. CCYV has also been shown to have very low genetic diversity globally based on the RNA-dependent RNA polymerase (RNA1), coat protein, and minor coat protein (RNA2) sequences [57]. The isolates from Georgia are closely related to isolates from many areas of the world, which make it difficult to determine the original source of introduction other than the fact that most likely these were introduced from other locations within the US, such as from neighboring states.

We also analyzed nucleic acid extracts from cantaloupe, cucumber, and squash by high throughput sequencing that can reveal the presence of novel or unsuspected agents along with already known agents [16,71,72]. Samples showing prominent symptoms on these crops were sequenced. No viruses other than CuLCrV, CCYV, and CYSDV were detected (Figure 2), underscoring the fact that these three viruses are the most important viruses affecting cucurbit production in Georgia during the fall. SqVYV, a whitefly transmitted *Ipomovirus* that causes a rapid vine decline in watermelon near the harvest, was also not detected. SqVYV is widely distributed in the southwest and west-central parts of Florida [73,74] and has also been detected in South Carolina [75], two neighboring states of Georgia, as well as California [76].

After the fall cucurbits are harvested in Georgia, there is a cucurbit-free period during the winter months before spring planting of watermelons begins in March. During that time, whitefly populations move from fall cucurbit plants to other cool-season crops, many of which could be hosts for CYSDV and CCYV. Many weed hosts of CYSDV [47] and CCYV [54,55] are also commonly found in Georgia. Watermelon and cantaloupe are the major cucurbit crops grown during the spring in Georgia. Additional surveys on winter and spring crops should provide a clearer picture of where the viruses overwinter in the state. The results from this study are also the first report of CCYV naturally infecting cantaloupe, cucumber, and zucchini in Georgia, USA.

## Figures and Tables

**Figure 1 viruses-13-00988-f001:**
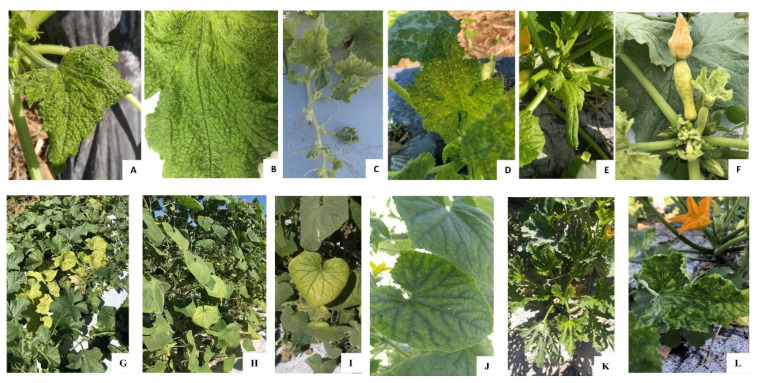
The different types of symptoms observed on squash plants included (**A**) chlorosis and crinkling, (**B**) vein yellowing, (**C**) interveinal leaf chlorosis, (**D**) yellow spots, and (**E**) severe stunting or bunching of leaves at the top of the plant. (**F**) Fruits, on squash plants, displayed severe bunching, were distorted, and were streaked with green patches. The main symptom observed on cantaloupe and cucumber was (**G**) interveinal leaf chlorosis, more prominent at the (**G**–**I**) crown region and milder on the (**J**) younger leaves. Foliage symptoms on zucchini including (**K**) crinkled leaves and a (**K**–**L**) yellow mosaic pattern.

**Figure 2 viruses-13-00988-f002:**
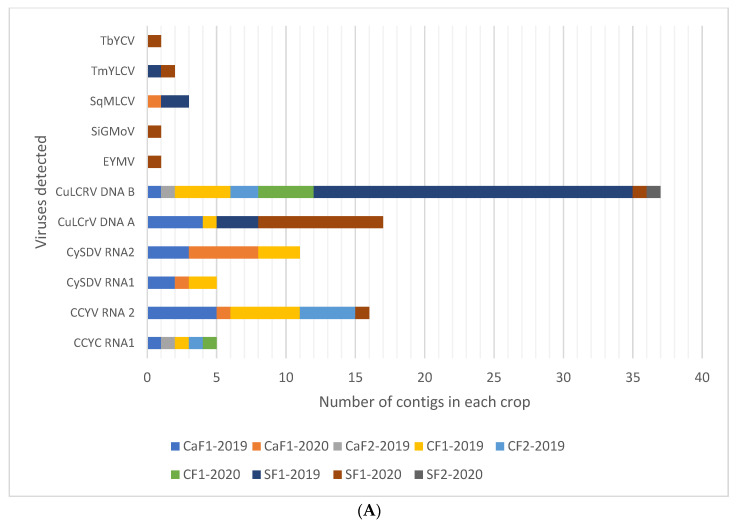
(**A**) Viruses detected on cantaloupe (CaF1-2019, CaF1-2020, CaF2-2019), cucumber (CF1-2019, CF2-2019, CF1-2020), and squash (SF1-2019, SF1-2020, SF2-2020) in Georgia by high throughput sequencing (HTS) of small RNAs (**A**) and RCA-HTS (**B**). Viruses detected and their genomic components are presented along the Y-axis. The number of contigs of each genome detected are presented along the X-axis. Each grid in the graph represents one contig and each color represents a different sample. Abbreviations used for viruses: Cucurbit chlorotic yellows virus (CCYV), cucurbit yellow stunting virus (CYSDV), cucurbit leaf crumple virus (CuLCrV), euphorbia yellow mosaic virus (EYMV), squash mild leaf curl virus (SqMLCV), sida golden mottle virus (SiGMoV), tomato mild yellow leaf curl virus (TmYLCV), and tobacco yellow crinkle virus (TbYCV).

**Figure 3 viruses-13-00988-f003:**
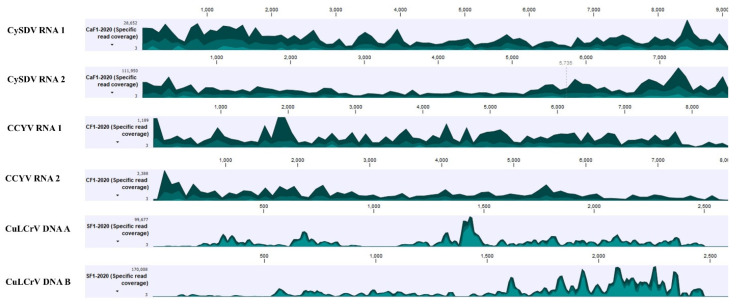
Read coverage maps of the viruses detected by high throughput sequencing of small RNAs of symptomatic cucurbits from Georgia. Scaled genome positions of the virus are shown above the histograms and the Y-axis represents the coverage in number of reads. Within the specified aggregation bucket, from top to bottom, the colors mean: The maximum coverage value (read count), the average coverage value, and the minimum coverage value. Abbreviations used for the viruses: Cucurbit chlorotic yellows virus (CCYV), cucurbit yellow stunting virus (CYSDV), cucurbit leaf crumple virus (CuLCrV), euphorbia yellow mosaic virus (EYMV), squash mild leaf curl virus (SqMLCV), sida golden mottle virus (SiGMoV), and tomato mild yellow leaf curl virus (TmYLCV).

**Figure 4 viruses-13-00988-f004:**
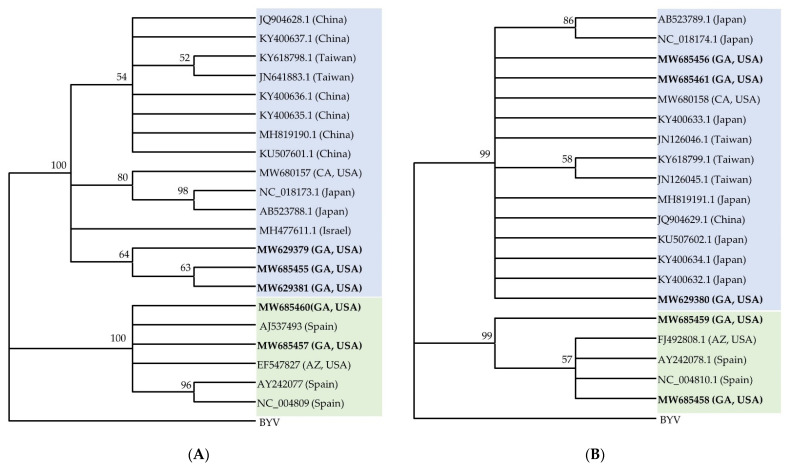
Phylogenetic relationship of coding regions of (**A**) RNA 1 and (**B**) RNA 2 of cucurbit chlorotic yellows virus (CCYV) and cucurbit yellow stunting disorder virus (CYSDV) isolates from Georgia with the corresponding sequences of CCYV and CYSDV isolates available in the GenBank. The country of origin of the sequences used are shown in parentheses after the accession numbers of viruses. CCYV sequences are shaded in blue and CYSDV sequences are in green. The Beet yellows virus (BYV) was set as an outgroup. Construction of the phylogenetic trees were done by the neighbor-joining method. The number on each node shows the percentage of bootstrap values (2000 replicates) in which a given node was recovered. Nodes with lower than 50 bootstrap values were collapsed since they were insignificant.

**Table 1 viruses-13-00988-t001:** Primers used for detection of viruses in this study.

Primer Name	Sequence 5′-3′	Tm (°C)	Amplicon Size	References
CCYV_RDRP_1515	CTCCGAGTAGATCATCCCAAATC	62	953	[6]
CCYV_RDRP_1515	TCACCAGAAACTCCACAATCTC			
CYSDV_RDRP_1542	TTTCGGCTCCCAGAGTTAATG	62	492	[28]
CYSDV_RDRP_1542	CGATCTCCGTGGTGTGATAAG			
CuLCRV CP 259 F	TCAAAGGTTTCCCGCTCTGC	58	588	(This study)
CuLCRV CP 846 R	TCCTGCTTCCTGGTGGTTGTAG			

**Table 2 viruses-13-00988-t002:** Incidence of begomovirus and criniviruses on fall cucurbits in Georgia in 2019 and 2020. The number of samples in which a virus was detected and their percentages (in parenthesis) are presented. The numbers for the virus detected at the highest frequency on a crop in a particular year are shown in bold.

	Virus ^a^	Cantaloupe		Cucumber		Squash		Zucchini	Total Number of Samples
		2019	2020	2019	2020	2019	2020	2020	
Single infections	CuLCrV	6 (29)	21 (53)	36 (53)	66 (69)	**195 (92)**	**274 (85)**	**52 (87)**	**650 (76)**
	CCYV	**21 (100)**	20 (50)	**53 (78)**	71 (74)	38 (18)	263 (82)	23 (38)	**489 (60)**
	CYSDV	12 (57)	**36 (90)**	6 (9)	**78 (81)**	31 (15)	161 (50)	25 (42)	**349 (43)**
Mixed infections	CuLCrV + CCYV	6 (29)	10 (25)	31 (46)	43 (45)	41 (19)	219 (68)	13 (22)	**363 (44)**
	CuLCrV + CySDV	4 (19)	14 (35)	2 (3)	49 (51)	27 (13)	154 (48)	18 (30)	**268 (33)**
	CYSDV + CCYV	12 (57)	18 (45)	6 (9)	58 (60)	9 (0.5)	128 (40)	17 (28)	**248 (30)**
	CuLCrV + CYSDV + CCYV	4 (19)	9 (23)	2 (3)	32 (33)	9 (0.5)	122 (38)	13 (21)	**191 (23)**
	Total number of samples tested	21	40	68	96	213	322	60	820

^a^ Virus acronyms used: Cucurbit leaf crumple virus (CuLCrV), cucurbit yellow stunting disorder virus (CYSDV), and cucurbit chlorotic yellows virus (CCYV). All samples collected in 2020 were also tested for cucurbit aphid borne yellows virus (CABYV), squash vein yellowing virus (SqVYV), and sida golden mosaic virus (SiGMV) but were not detected in any of the samples.

## Data Availability

Not applicable.

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
