# Peer review of "High Throughput Sequencing-Aided Survey Reveals Widespread Mixed Infections of Whitefly-Transmitted Viruses in Cucurbits in Georgia, USA"

_viruses, 2021, doi:10.3390/v13060988_

Round 1
Reviewer 1 Report
The manuscript Viruses-1148822 by Kavalappara et al. reports on a survey of whitefly-transmitted viruses of cucurbits aided by HTS analysis in Georgia. The manuscript is written generally very well and has publishable results on virus survey in cucurbits which would be of significance to the region. However, data presented here do not add much to the biology or ecology of WFT-viruses and I am skeptical on the interest of readers of Viruses to this manuscript. I believe this is a decision that needs to be made by the Editor/Sr. Editor.
The manuscript needs a revision to address the comments and edits listed below before it can be accepted for publication.
- I would have liked to see data on economic impact due to WFT viruses. Ref (2) provided is anecdotal not peer reviewed. Can the authors provide other peer reviewed references?
- As high as 75% incidence of CuLCrV has been reported in snap beans from GA (Ref 3; Check author list for this reference). The high incidence of WFT-viruses is not surprising. It would be nice if authors can include a map of GA with the location of counties where the samples were collected and WFT-viruses of cucurbits were found.
- Authors have relied on data from small RNA sequencing. Researchers have used dsRNA, RNAseq and small RNA sequencing over the last 12 years to discover the viruses. No single method is proved to be better than the others. However, small RNA profile can be influenced by silencing suppressors encoded by viruses. I am not sure if this is the case where low contigs/reads were found for 5 begomoviruses in Figure 2 and 3. An RCA-based confirmation of these viruses in samples would be able to confirm the virus presence even if the PCR reaction fail because of poor sensitivity of primers not being sensitive due to genetic diversity in these viruses.
- The HTS reads from small RNA sequencing may allow assembly of chimeras. Genomes of viruses need to be verified by Sanger Sequencing and complete genome sequences deposited to NCBI. Phylogeny developed using sequences from Illumina reads are not reliable.
- Figure 1 has an array of symptoms for different hosts. However, it is not clear what the symptoms would be if the viruses present were one, two or three, of CCYV, CuLCrV and CYSDV.
- In Figure 2, the poor representation in reads for five begomoviruses, makes me wonder if this was an artifact of sequencing or time of sampling when the plants might have very low small RNA profile. Based on Table 1, only three viruses were verified by RT-PCR or PCR.
- In line 197, authors claim detection of sequences of other begomoviruses. Are there any PCR results to confirm if they were present in the samples or not?
- Table 2 would be useful, if the break-up of reads for each virus discovered is given as % of total reads. In the present state, it can be moved to supplementary information.
Minor edits:
Table 3, 5, and Supplementary Table 1. List 2019 before 2020.
Supplementary Table 1. Expand the virus acronyms.
Line 48 and 49. Addition of information on tospoviruses is not in line with the title. Suggest deletion.
Line 50 . …neighboring state of Florida where high incidence…
Line 52, 89 Use ‘High throughput sequencing (HTS)’ as opposed to NGS. Term NGS is outdated with many more next generation sequencing methods being around.
Line 75 counties,
Line 85 to 88 Expand BGI before abbreviating it. Be consistent in providing the company, City, State and Country. See lines 88, 117 and 120. Add a comma after CA in CA USA.
Line 90 virus detection and identification in…
Line 107-110 This method must have been published, provide a citation.
Line 123 Provide source of random hexamers.
Line 130, 212, 239, and… Avoid starting sentences with acronyms.
Line 168 as well as disease incidence … (incidence is actually rate).
Line 225 .. at high frequency… (or higher than…)
Line 234 provide length of the sequence
Line 258-260 This text is already covered at the beginning of introduction. Delete
Line 263 Replaced ‘expected’ by ‘believed’
Line 267 … persistently-transmitted…
Line 330 …alternative hosts… (alternate hosts are those needed to complete life cycle as in wheat rust and barberry bushes)
Line 362 …cucurbits… or cucurbit crops
Line 639 Number of contigs of….
Line 674 In Figure (a) is (A) and (b) is (B). Fix.
Line 688&689 Formatting problem!
Line 711 Indicate if the sequences were obtained from RT-PCR products in the caption.
Author Response
Our response to the reviewers (in blue).
Reviewer one
- I would have liked to see data on economic impact due to WFT viruses. Ref (2) provided is anecdotal not peer reviewed. Can the authors provide other peer reviewed references?
Response: There are no peer-reviewed references on economic impact due to WTVs in Georgia.
- As high as 75% incidence of CuLCrV has been reported in snap beans from GA (Ref 3; Check author list for this reference). The high incidence of WFT-viruses is not surprising. It would be nice if authors can include a map of GA with the location of counties where the samples were collected and WFT-viruses of cucurbits were found.
Response: A map of Georgia with the locations of counties has been added as a supplementary figure 1.
- Authors have relied on data from small RNA sequencing. Researchers have used dsRNA, RNAseq and small RNA sequencing over the last 12 years to discover the viruses. No single method is proved to be better than the others. However, small RNA profile can be influenced by silencing suppressors encoded by viruses. I am not sure if this is the case where low contigs/reads were found for 5 begomoviruses in Figure 2 and 3. An RCA-based confirmation of these viruses in samples would be able to confirm the virus presence even if the PCR reaction fail because of poor sensitivity of primers not being sensitive due to genetic diversity in these viruses.
Response: We have included results from high throughput sequencing of RCA products of cucurbit samples collected in 2019. It was not included with the earlier draft since no virus other than CuLCrV was detected. (Line 140-147 and Lines 206-209)
- The HTS reads from small RNA sequencing may allow assembly of chimeras. Genomes of viruses need to be verified by Sanger Sequencing and complete genome sequences deposited to NCBI. Phylogeny developed using sequences from Illumina reads are not reliable.
Response: There are many recent articles where the sequences from Illumina reads have been used to ascertain phylogeny. The presence of the virus has been confirmed from the test samples Sanger sequencing of the target genes. Below are recent examples published in Viruses and Plant Disease:
- Bornancini, V.A.; Irazoqui, J.M.; Flores, C.R.; Vaghi Medina, C.G.; Amadio, A.F.; López Lambertini, P.M. Reconstruction and Characterization of Full-Length Begomovirus and Alphasatellite Genomes Infecting Pepper through Metagenomics. Viruses2020, 12, 202. https://doi.org/10.3390/v12020202
- Vučurović, A.; Kutnjak, D.; Mehle, N.; Stanković, I.; Pecman, A.; Bulajić, A.; Krstić, B.; Ravnikar, M. Detection of Four New Tomato Viruses in Serbia using Post-Hoc High-Throughput Sequencing Analysis of Samples from a Large-Scale Field Survey. 2021 Plant Dis. 24 Mar 2021 https://doi.org/10.1094/PDIS-09-20-1915-RE
- Figure 1 has an array of symptoms for different hosts. However, it is not clear what the symptoms would be if the viruses present were one, two or three, of CCYV, CuLCrV and CYSDV.
Response: It is difficult to differentiate symptoms of each virus unless they are done in controlled greenhouse conditions. However, we believe these figures will help our readers, especially field plant virologists, all over the world in associating these symptoms with any of the three viruses. Often times, as we have mentioned in the article, if one virus is consistently detected, the others go unnoticed and could lead to failure of management efforts. However, we acknowledge that we cannot say with confidence which plants are infected with which virus. We request you to reconsider.
- In Figure 2, the poor representation in reads for five begomoviruses, makes me wonder if this was an artifact of sequencing or time of sampling when the plants might have very low small RNA profile. Based on Table 1, only three viruses were verified by RT-PCR or PCR.
Response: As mentioned in response to comment No.3, the presence for these viruses were checked for using Rolling Circle Amplification and none were found. (Line 140-147 and Lines 206-209).
- In line 197, authors claim detection of sequences of other begomoviruses. Are there any PCR results to confirm if they were present in the samples or not?
Response: We have later ruled out their presence both by low coverage of reads as well as by sequencing the RCA products. (Line 140-147 and Lines 206-209).
- Table 2 would be useful, if the break-up of reads for each virus discovered is given as % of total reads. In the present state, it can be moved to supplementary information.
Response: This table has been moved to supplementary information (supplementary Table 1).
Minor edits:
Table 3, 5, and Supplementary Table 1. List 2019 before 2020.
The tables have been modified as suggested
Supplementary Table 1. Expand the virus acronyms.
The virus acronyms have been expanded.
Line 48 and 49. Addition of information on tospoviruses is not in line with the title. Suggest deletion.
This information has been deleted.
Line 50 . …neighboring state of Florida where high incidence…
The sentence has been modified as “a potential threat to cucurbits and melons [8] in Georgia due to its presence in neighboring state of Florida”.
Line 52, 89 Use ‘High throughput sequencing (HTS)’ as opposed to NGS. Term NGS is outdated with many more next generation sequencing methods being around.
Term HTS is used in place of NGS as suggested.
Line 75 counties,
Modified as suggested
Line 85 to 88 Expand BGI before abbreviating it. Be consistent in providing the company, City, State and Country. See lines 88, 117 and 120. Add a comma after CA in CA USA.
Modified as suggested
Line 90 virus detection and identification in…
The line is modified as “Small RNA processing, assembly, virus detection and identification in the samples…”
Line 107-110 This method must have been published, provide a citation.
The method was adopted from the manufacturer’s instruction, with modification done in our lab, including use of 4M Guanidine isothiocyanate (GITC) buffer (pH 5.0) in place of buffer provided with kit for sample homogenization by mechanical disruption. The procedure is explained in detail so as to make it easier for other labs to adopt.
Line 123 Provide source of random hexamers.
The source “(Thermo Fisher Scientific, USA)” is included in the text.
Line 130, 212, 239, and… Avoid starting sentences with acronyms.
This is a good suggestion, and we tried to change as much as possible.
Line 168 as well as disease incidence … (incidence is actually rate).
Line 225 .. at high frequency… (or higher than…)
The viruses were detected in higher frequency
Line 234 provide length of the sequence
The sequence length has been included in the text. Line 252-254; 262-264.
Line 258-260 This text is already covered at the beginning of introduction. Delete
The text is deleted
Line 263 Replaced ‘expected’ by ‘believed’
The text is modified as suggested
Line 267 … persistently-transmitted…
Modified as suggested
Line 330 …alternative hosts… (alternate hosts are those needed to complete life cycle as in wheat rust and barberry bushes)
Modified as suggested
Line 362 …cucurbits… or cucurbit crops
Modified as suggested
Line 639 Number of contigs of….
Modified as suggested
Line 674 In Figure (a) is (A) and (b) is (B). Fix.
Modified as suggested
Line 688&689 Formatting problem!
Could not understand the comment.
Line 711 Indicate if the sequences were obtained from RT-PCR products in the caption.
The caption is modified as “Assembled from siRNA sequences”
Reviewer 2 Report
Dear colleagues,
This review is concerning a research work entitled “ High throughput sequencing-aided survey reveals widespread mixed infections of whitefly transmitted viruses in cucurbits in Georgia USA, by Saritha Raman Kavalappara, Hayley Milner, Naga Charan Konakalla, Kaelyn Morgan, Alton Sparks, Cecilia McGregor, Albert K. Culbreath, William M. Wintermantel and Sudeep Bag. As detailed experiments I recommend it for an international audience in this journal, however several points have to be precised and a major revision is requested.
Please notice that the three major points of my comments (at the beginning) are very important (mandatory…) for a suitable value and understanding of the article, for specialists and non-specialists, in order to bring a broader audience in this journal. Minor points are also enhanced at the end of this review.
I deeply hope to see this article published soon,
The three major point are:
- 1-As this paper concerns a disease appearing or not in various taxa of Cucurbitaceae, even on a large computer screen the photos of figure 1 are too small. In the present manuscript almost no detail is visible and it is quite frustrating for all these very interesting set of observations centered on viruses disease. The two sets of symptoms detailed in 3.1 should be divided in two figures with higher enlargements; moreover, for each photo the caption should detail precisely which virus(es) is involved. Finally sections of leaf (or other organs) are highly suitable to see the damage caused by the virus(es) in the material (cells, tissues…). Transmission microscopy or other type of microscope would be welcome, but this is probably more difficult to get than Light microscope routinely used in laboratories.
- 2-Checking briefly in the word of science WOS with key-words and topics like “cantaloupe, cucumber, yellow squash, chlorotic yellows virus (CCYV), cucurbit leaf crumple virus (CuLCrV), cucurbit yellow stunting disorder virus (CYSDV)”…, much more articles (very recent from 2021 or a little bit “older”) appear and they should be once more selected (or “older” ones…) and used in their details and values provided (not just citing them) to increase the quality of the discussion for an international audience in this journal. Among these are the 22 following references:
- Chang, H.Y.; Chen, L.C.; Lin, C.C.; Tsai, W.S. First report of cucurbit chlorotic yellows virus infecting melon, watermelon and wild melon in the Philippines. J Plant Pathol 2021, 10.1007/s42161-021-00768-7, doi:10.1007/s42161-021-00768-7.
- Agarwal, G.; Kavalappara, S.R.; Gautam, S.; da Silva, A.; Simmons, A.; Srinivasan, R.; Dutta, B. Field Screen and Genotyping of Phaseolus vulgaris against Two Begomoviruses in Georgia, USA. Insects 2021, 12, doi:ARTN 49 10.3390/insects12010049.
- Waliullah, S.; Ling, K.S.; Cieniewicz, E.J.; Oliver, J.E.; Ji, P.S.; Ali, M.E. Development of Loop-Mediated Isothermal Amplification Assay for Rapid Detection of Cucurbit Leaf Crumple Virus. Int J Mol Sci 2020, 21, doi:ARTN 1756 10.3390/ijms21051756.
- Topkaya, S.; Desbiez, C. Molecular characterization of Cucurbit aphid-borne yellows virus (CABYV) affecting cucurbits in Turkey. Zemdirbyste 2020, 107, 353-358, doi:10.13080/z-a.2020.107.045.
- Salavert, F.; Navarro, J.A.; Owen, C.A.; Khechmar, S.; Pallas, V.; Livieratos, I.C. Cucurbit chlorotic yellows virus p22 suppressor of RNA silencing binds single-, double-stranded long and short interfering RNA molecules in vitro. Virus Res 2020, 279, doi:ARTN 197887 10.1016/j.virusres.2020.197887.
- Perez-de-Castro, A.; Lopez-Martin, M.; Esteras, C.; Garces-Claver, A.; Palomares-Rius, F.J.; Pico, M.B.; Gomez-Guillamon, M.L. Melon Genome Regions Associated with TGR-1551-Derived Resistance toCucurbit yellow stunting disorder virus. Int J Mol Sci 2020, 21, doi:ARTN 5970 10.3390/ijms21175970.
- Kheireddine, A.; Saez, C.; Sifres, A.; Pico, B.; Lopez, C. First Report of Cucurbit Chlorotic Yellows Virus Infecting Cucumber and Zucchini in Algeria. Plant Dis 2020, 104, 1264-1265, doi:10.1094/Pdis-10-19-2091-Pdn.
- Kalischuk, M.L.; Roberts, P.D.; Paret, M.L. A rapid fluorescence-based real-time isothermal assay for the detection of Cucurbit yellow stunting disorder virus in squash and watermelon plants. Mol Cell Probe 2020, 53, doi:ARTN 101613 10.1016/j.mcp.2020.101613.
- Kalischuk, M.; Meru, G.; Paret, M. A Rapid Isothermal Assay for the Detection of Cucurbit Leaf Crumple Virus in Watermelon. Hortscience 2020, 55, S136-S137.
- Gautam, S.; Gadhave, K.R.; Buck, J.W.; Dutta, B.; Coolong, T.; Adkins, S.; Srinivasan, R. Virus-virus interactions in a plant host and in a hemipteran vector: Implications for vector fitness and virus epidemics. Virus Res 2020, 286, doi:ARTN 198069 10.1016/j.virusres.2020.198069.
- Gadhave, K.R.; Gautam, S.; Dutta, B.; Coolong, T.; Adkins, S.; Srinivasan, R. Low Frequency of Horizontal and Vertical Transmission of Cucurbit Leaf Crumple Virus in Whitefly Bemisia tabaci Gennadius. Phytopathology 2020, 110, 1235-1241, doi:10.1094/Phyto-09-19-0337-R.
- Domingo-Calap, M.L.; Moreno, A.B.; Pendon, J.A.D.; Moreno, A.; Fereres, A.; Lopez-Moya, J.J. Assessing the Impact on Virus Transmission and Insect Vector Behavior of a Viral Mixed Infection in Melon. Phytopathology 2020, 110, 174-186, doi:10.1094/Phyto-04-19-0126-Fi.
- Desbiez, C.; Wipf-Scheibel, C.; Millot, P.; Berthier, K.; Girardot, G.; Gognalons, P.; Hirsch, J.; Moury, B.; Nozeran, K.; Piry, S., et al. Distribution and evolution of the major viruses infecting cucurbitaceous and solanaceous crops in the French Mediterranean area. Virus Res 2020, 286, doi:ARTN 198042 10.1016/j.virusres.2020.198042.
- Ahmad, M.H.; Al-Saleh, M.A.; Al-Shahwan, I.M.; Shakeel, M.T.; Ibrahim, Y.E.; Amer, M.A. Characterization of Cucurbit Yellow Stunting Disorder Virus Associated with Yellowing Disease of Watermelon in Saudi Arabia. J Anim Plant Sci-Pak 2020, 30, 1206-1214, doi:10.36899/Japs.2020.5.0138.
- Wei, Y.; Shi, Y.J.; Han, X.Y.; Chen, S.Y.; Li, H.L.; Chen, L.L.; Sun, B.J.; Shi, Y. Identification of cucurbit chlorotic yellows virus P4.9 as a possible movement protein. Virol J 2019, 16, doi:ARTN 82 10.1186/s12985-019-1192-y.
- Wang, Y.; Zhu, P.; Zhou, Q.; Zhou, X.J.; Guo, Z.Q.; Cheng, L.R.; Zhu, L.Y.; He, X.C.; Zhu, Y.D.; Hu, Y. Detection of disease in Cucurbita maxima Duch. ex Lam. caused by a mixed infection of Zucchini yellow mosaic virus, Watermelon mosaic virus, and Cucumber mosaic virus in Southeast China using a novel small RNA sequencing method. Peerj 2019, 7, doi:ARTN e7930 10.7717/peerj.7930.
- Orfanidou, C.G.; Mathioudakis, M.M.; Katsarou, K.; Livieratos, I.; Katis, N.; Maliogka, V.I. Cucurbit chlorotic yellows virus p22 is a suppressor of local RNA silencing. Archives of Virology 2019, 164, 2747-2759, doi:10.1007/s00705-019-04391-x.
- Lu, S.H.; Chen, M.S.; Li, J.J.; Shi, Y.; Gu, Q.S.; Yan, F.M. Changes in Bemisia tabaci feeding behaviors caused directly and indirectly by cucurbit chlorotic yellows virus. Virol J 2019, 16, doi:ARTN 106 10.1186/s12985-019-1215-8.
- Supakitthanakorn, S.; Akarapisan, A.; Ruangwong, O.U. First record of melon yellow spot virus in pumpkin and its occurrence in cucurbitaceous crops in Thailand. Australas Plant Dis 2018, 13, doi:ARTN 32 10.1007/s13314-018-0314-5.
- Coolong, T.; Dutta, B.; Sparks, A.; Srinivasan, R. Evaluation of Production Practices for Tolerance to Cucurbit Leaf Crumple Virus In Yellow Squash and Zucchini. Hortscience 2018, 53, S477-S477.
- Castle, S.; Palumbo, J.; Merten, P.; Cowden, C.; Prabhaker, N. Effects of foliar and systemic insecticides on whitefly transmission and incidence of Cucurbit yellow stunting disorder virus. Pest Manag Sci 2017, 73, 1462-1472, doi:10.1002/ps.4478.
- Carriere, Y.; Degain, B.; Hartfield, K.A.; Nolte, K.D.; Marsh, S.E.; Ellers-Kirk, C.; Van Leeuwen, W.J.D.; Liesner, L.; Dutilleul, P.; Palumbo, J.C. Assessing Transmission of Crop Diseases by Insect Vectors in a Landscape Context. J Econ Entomol 2014, 107, 1-10, doi:10.1603/Ec13362.
- 3- The main point in the present state of the manuscript concerns the discussion part which is more a synthesis of the results than a real discussion and it has to be much more developed, emphasizing your interesting data. The relation between bibliographic data and the interest of the present data is not enough explored, it is in some paragraphs very difficult to state if the authors are reporting about their material or about the bibliographical data. Among possibilities it would be extremely welcome to discuss the phylogenetic trees as they are in many papers, proposing hypotheses for instance in terms of comparisons between “sister groups” or not: according to the figures provided, trees A and B are different, some USA viruses are grouped differently and some show a "proximity" or not with other worldwide origins, remarks allowing propositions for the future as the manuscript is about a disease... As details, would the first paragraph of this part be more suitable in the introduction part? Please use bibliographic data (above and others…) in all their details (values of tables, etc…) for the improvement of all this part.
As minor points:
1 the title seems quite long, usually a short title is much better memorized by readers, may be "High throughput sequencing-aided survey reveals" could be deleted?
2 in the abstract, what does "n" mean? (= haploid number of chromosomes?) As apparently you do not use this in the text, is it necessary? Moreover, normally you should use this only if you checked by yourself the number of chromosomes (?), and as you experimented with commercial material you cannot be sure at all of the number of chromosomes without checking them by observation of mitosis cells… [believe me, polyploidy is very (extremely…) frequent in agronomy selections]; homogenize with points 4 and 5 just below;
3 in the introduction, as “Cucurbitaceae” is already a family taxonomical level (ending with “…ceae”), you cannot use it with “family” together, put” family” somewhere else in the sentence; change this also in the discussion line 274 (“in the family Cucurbitaceae and Fabaceae”);
4 in the introduction, put (at least once in the text) the name(s) or abbreviation(s) of the author(s) of latin names of the plants (use international plant names index IPNI internet or another relevant up-to-date site); do the same for the viruses as you use or not details of genus taxonomical rank and it has to be homogeneous; precise also that Bemicia tabaci is an insect; precise also what are “zucchini and beans”, for non-specialist it will be very helpful; homogenize with points 2 and 5 just after;
5 related with point 4 just above, in material and methods (or in the introduction) precise if the plants are cultivars, subspecies… it may influence your discussion if your collection is heterogeneous and/or not for one single taxon; cultivars are not wild taxon, they may have different chromosome numbers; homogenize with points 2 and 4 just before;
6 in the results 3.1, are all these symptoms in the figure 1?
7 in the discussion line 295, is “Crinvirus” correct?
8 in the discussion, precise if from the paragraph “Most damage due to WTVs is…” it concerns your experiments as it is not enough clear;
9 in the discussion line 328, what is WFT virus? For non-specialist it has to be clarified;
10 for figure 2, add a horizontal title for the plants and a vertical title for viruses, it will be more rapid to check;
11 for figure 3, the quality of the figures is not good at all, we cannot read the letters; may be it is better to divide the data in two figures in order to enlarge each set of data;
12 for figure 4, use colours to enhance your experimented material.
Author Response
The three major point are:
1-As this paper concerns a disease appearing or not in various taxa of Cucurbitaceae, even on a large computer screen the photos of figure 1 are too small. In the present manuscript almost no detail is visible and it is quite frustrating for all these very interesting set of observations centered on viruses disease. The two sets of symptoms detailed in 3.1 should be divided in two figures with higher enlargements; moreover, for each photo the caption should detail precisely which virus(es) is involved. Finally sections of leaf (or other organs) are highly suitable to see the damage caused by the virus(es) in the material (cells, tissues…). Transmission microscopy or other type of microscope would be welcome, but this is probably more difficult to get than Light microscope routinely used in laboratories.
- Response: It is difficult to differentiate symptoms of each virus unless they are done in controlled greenhouse conditions. However, we believe these figures will help our readers, especially field plant virologists, all over the world in associating these symptoms with any of the three viruses mentioned in our article. Often times, as we have mentioned in the article, if one virus is consistently detected, the others go unnoticed and could lead to failure of management efforts.
- 2-Checking briefly in the word of science WOS with key-words and topics like “cantaloupe, cucumber, yellow squash, chlorotic yellows virus (CCYV), cucurbit leaf crumple virus (CuLCrV), cucurbit yellow stunting disorder virus (CYSDV)”…, much more articles (very recent from 2021 or a little bit “older”) appear and they should be once more selected (or “older” ones…) and used in their details and values provided (not just citing them) to increase the quality of the discussion for an international audience in this journal. Among these are the 22 following references:
- Chang, H.Y.; Chen, L.C.; Lin, C.C.; Tsai, W.S. First report of cucurbit chlorotic yellows virus infecting melon, watermelon and wild melon in the Philippines. J Plant Pathol 2021, 10.1007/s42161-021-00768-7, doi:10.1007/s42161-021-00768-7.
- Agarwal, G.; Kavalappara, S.R.; Gautam, S.; da Silva, A.; Simmons, A.; Srinivasan, R.; Dutta, B. Field Screen and Genotyping of Phaseolus vulgaris against Two Begomoviruses in Georgia, USA. Insects 2021, 12, doi:ARTN 49 10.3390/insects12010049.
- Waliullah, S.; Ling, K.S.; Cieniewicz, E.J.; Oliver, J.E.; Ji, P.S.; Ali, M.E. Development of Loop-Mediated Isothermal Amplification Assay for Rapid Detection of Cucurbit Leaf Crumple Virus. Int J Mol Sci 2020, 21, doi:ARTN 1756 10.3390/ijms21051756.
- Topkaya, S.; Desbiez, C. Molecular characterization of Cucurbit aphid-borne yellows virus (CABYV) affecting cucurbits in Turkey. Zemdirbyste 2020, 107, 353-358, doi:10.13080/z-a.2020.107.045.
- Salavert, F.; Navarro, J.A.; Owen, C.A.; Khechmar, S.; Pallas, V.; Livieratos, I.C. Cucurbit chlorotic yellows virus p22 suppressor of RNA silencing binds single-, double-stranded long and short interfering RNA molecules in vitro. Virus Res 2020, 279, doi:ARTN 197887 10.1016/j.virusres.2020.197887.
- Perez-de-Castro, A.; Lopez-Martin, M.; Esteras, C.; Garces-Claver, A.; Palomares-Rius, F.J.; Pico, M.B.; Gomez-Guillamon, M.L. Melon Genome Regions Associated with TGR-1551-Derived Resistance toCucurbit yellow stunting disorder virus. Int J Mol Sci 2020, 21, doi:ARTN 5970 10.3390/ijms21175970.
- Kheireddine, A.; Saez, C.; Sifres, A.; Pico, B.; Lopez, C. First Report of Cucurbit Chlorotic Yellows Virus Infecting Cucumber and Zucchini in Algeria. Plant Dis 2020, 104, 1264-1265, doi:10.1094/Pdis-10-19-2091-Pdn.
- Kalischuk, M.L.; Roberts, P.D.; Paret, M.L. A rapid fluorescence-based real-time isothermal assay for the detection of Cucurbit yellow stunting disorder virus in squash and watermelon plants. Mol Cell Probe 2020, 53, doi:ARTN 101613 10.1016/j.mcp.2020.101613.
- Kalischuk, M.; Meru, G.; Paret, M. A Rapid Isothermal Assay for the Detection of Cucurbit Leaf Crumple Virus in Watermelon. Hortscience 2020, 55, S136-S137.
- Gautam, S.; Gadhave, K.R.; Buck, J.W.; Dutta, B.; Coolong, T.; Adkins, S.; Srinivasan, R. Virus-virus interactions in a plant host and in a hemipteran vector: Implications for vector fitness and virus epidemics. Virus Res 2020, 286, doi:ARTN 198069 10.1016/j.virusres.2020.198069.
- Gadhave, K.R.; Gautam, S.; Dutta, B.; Coolong, T.; Adkins, S.; Srinivasan, R. Low Frequency of Horizontal and Vertical Transmission of Cucurbit Leaf Crumple Virus in Whitefly Bemisia tabaci Gennadius. Phytopathology 2020, 110, 1235-1241, doi:10.1094/Phyto-09-19-0337-R.
- Domingo-Calap, M.L.; Moreno, A.B.; Pendon, J.A.D.; Moreno, A.; Fereres, A.; Lopez-Moya, J.J. Assessing the Impact on Virus Transmission and Insect Vector Behavior of a Viral Mixed Infection in Melon. Phytopathology 2020, 110, 174-186, doi:10.1094/Phyto-04-19-0126-Fi.
- Desbiez, C.; Wipf-Scheibel, C.; Millot, P.; Berthier, K.; Girardot, G.; Gognalons, P.; Hirsch, J.; Moury, B.; Nozeran, K.; Piry, S., et al. Distribution and evolution of the major viruses infecting cucurbitaceous and solanaceous crops in the French Mediterranean area. Virus Res 2020, 286, doi:ARTN 198042 10.1016/j.virusres.2020.198042.
- Ahmad, M.H.; Al-Saleh, M.A.; Al-Shahwan, I.M.; Shakeel, M.T.; Ibrahim, Y.E.; Amer, M.A. Characterization of Cucurbit Yellow Stunting Disorder Virus Associated with Yellowing Disease of Watermelon in Saudi Arabia. J Anim Plant Sci-Pak 2020, 30, 1206-1214, doi:10.36899/Japs.2020.5.0138.
- Wei, Y.; Shi, Y.J.; Han, X.Y.; Chen, S.Y.; Li, H.L.; Chen, L.L.; Sun, B.J.; Shi, Y. Identification of cucurbit chlorotic yellows virus P4.9 as a possible movement protein. Virol J 2019, 16, doi:ARTN 82 10.1186/s12985-019-1192-y.
- Wang, Y.; Zhu, P.; Zhou, Q.; Zhou, X.J.; Guo, Z.Q.; Cheng, L.R.; Zhu, L.Y.; He, X.C.; Zhu, Y.D.; Hu, Y. Detection of disease in Cucurbita maxima Duch. ex Lam. caused by a mixed infection of Zucchini yellow mosaic virus, Watermelon mosaic virus, and Cucumber mosaic virus in Southeast China using a novel small RNA sequencing method. Peerj 2019, 7, doi:ARTN e7930 10.7717/peerj.7930.
- Orfanidou, C.G.; Mathioudakis, M.M.; Katsarou, K.; Livieratos, I.; Katis, N.; Maliogka, V.I. Cucurbit chlorotic yellows virus p22 is a suppressor of local RNA silencing. Archives of Virology 2019, 164, 2747-2759, doi:10.1007/s00705-019-04391-x.
- Lu, S.H.; Chen, M.S.; Li, J.J.; Shi, Y.; Gu, Q.S.; Yan, F.M. Changes in Bemisia tabaci feeding behaviors caused directly and indirectly by cucurbit chlorotic yellows virus. Virol J 2019, 16, doi:ARTN 106 10.1186/s12985-019-1215-8.
- Supakitthanakorn, S.; Akarapisan, A.; Ruangwong, O.U. First record of melon yellow spot virus in pumpkin and its occurrence in cucurbitaceous crops in Thailand. Australas Plant Dis 2018, 13, doi:ARTN 32 10.1007/s13314-018-0314-5.
- Coolong, T.; Dutta, B.; Sparks, A.; Srinivasan, R. Evaluation of Production Practices for Tolerance to Cucurbit Leaf Crumple Virus In Yellow Squash and Zucchini. Hortscience 2018, 53, S477-S477.
- Castle, S.; Palumbo, J.; Merten, P.; Cowden, C.; Prabhaker, N. Effects of foliar and systemic insecticides on whitefly transmission and incidence of Cucurbit yellow stunting disorder virus. Pest Manag Sci 2017, 73, 1462-1472, doi:10.1002/ps.4478.
- Carriere, Y.; Degain, B.; Hartfield, K.A.; Nolte, K.D.; Marsh, S.E.; Ellers-Kirk, C.; Van Leeuwen, W.J.D.; Liesner, L.; Dutilleul, P.; Palumbo, J.C. Assessing Transmission of Crop Diseases by Insect Vectors in a Landscape Context. J Econ Entomol 2014, 107, 1-10, doi:10.1603/Ec13362.
Response: These articles are all mostly good reads on their own about chlorotic yellows virus (CCYV), cucurbit leaf crumple virus (CuLCrV), and cucurbit yellow stunting disorder virus (CYSDV). We have included the findings from those articles which contribute toward explaining the high degree of single as well as mixed infections found in our study, including some of those listed by the reviewer as well as others we previously selected. However, the areas of research covered in many of these articles do not come under the scope of our study even though they talk about the same viruses. It is not appropriate to cite every recent publication, rather only those relevant to the research presented are included here.
Discussion included in the text:
Plant viruses influence the behavior and physiology of their vectors to increase their transmission. CCYV has been shown to affect the feeding behaviors of B. tabaci, which may increase the ability of its vector B. tabaci for CCYV transmission (Lu et al., 2019). In CCYV and CYSDV pathosystem on cucumber, nonviruliferous whiteflies preferentially settled on virus-infected host plants whereas viruliferous whiteflies were more attracted to healthy plants (Orfanidou et al 2021), thus making effective use of existing source of inoculum for initiating secondary spread. In contrast, in the yellow squash- viral pathosystem (CuLCrV and/or CYSDV), both non-viruliferous and viruliferous whiteflies preferred to settle on non-infected plants (Gadhave et al 2020). In southeastern United States, an overwhelming number of non-viruliferous whiteflies disperse from non-virus hosts (such as cotton) to vegetable farmscapes during fall, and acquire one or more squash viruses. Preferential settling of these viruliferous whiteflies on non-infected plants can then enhance virus spread of one or more viruses in the susceptible crops causing heavy incidence of these viruses (Gadhave et al 2020).
Virus-virus interactions associated with mixed infections can shape the population dynamics of component viruses (Wintermantel et al 2008, Palma et al, 2010, Gautam et al, 2020) in a host. A few studies have illustrated the crop-dependent preferential accumulation and transmission of one of the viruses among CCYV, CYSDV and CuLCrV over others during mixed infections. CYSDV accumulated in significantly lower amounts in yellow squash plants infected with both CCYV and CYSDV than those infected with only CYSDV. Whiteflies acquired similar levels of CuLCrV but reduced levels of CYSDV from mixed-infected squash plants in comparison to plants infected with only any one of these viruses (Gadhave et al 2020). This may partially explain the higher numbers of CuLCrV than any of the criniviruses which were observed on yellow squash in this survey in 2019 and 2020. In melon (cantaloupe), on the other hand, the rates of whitefly transmission of CYSDV increased when plants dually infected with CYSDV and watermelon mosaic virus (WMV), a potyvirus was used as source (Domingo et al 2020). In cucumber, the accumulation and as a result the transmission efficiencies of both CCYV and CYSDV, by whiteflies were significantly decreased during mixed infections, However, their simultaneous transmission efficiency was significantly higher (Orafanidu et al 2021). These However, their findings cannot completely explain the differences in virus frequencies which were observed on cantaloupe and yellow squash in this study.
3- The main point in the present state of the manuscript concerns the discussion part which is more a synthesis of the results than a real discussion and it has to be much more developed, emphasizing your interesting data. The relation between bibliographic data and the interest of the present data is not enough explored, it is in some paragraphs very difficult to state if the authors are reporting about their material or about the bibliographical data. Among possibilities it would be extremely welcome to discuss the phylogenetic trees as they are in many papers, proposing hypotheses for instance in terms of comparisons between “sister groups” or not: according to the figures provided, trees A and B are different, some USA viruses are grouped differently and some show a "proximity" or not with other worldwide origins, remarks allowing propositions for the future as the manuscript is about a disease... As details, would the first paragraph of this part be more suitable in the introduction part? Please use bibliographic data (above and others…) in all their details (values of tables, etc…) for the improvement of all this part.
Response: While talking about phylogeny tree it was a conscious decision to not delve into the details of phylogeny since these are all known viruses and it's already known what groups they are in. Furthermore, particularly for these viruses, there is a high degree of sequence identity among isolates from throughout the world, with the primary exception being CYSDV isolates from Saudi Arabia and connected locations which diverge from other CYSDV isolates. Our aim was to make the point that based on the information we have and that previously known, the isolates in Georgia are quite closely related to isolates from many areas of the world, which may make it difficult to determine the original source of introduction. Had sequences been more divergent, it might suggest a mode of introduction or possible place from which they moved to Georgia, but with the high sequence identities it may be difficult to determine an origin other than most likely these isolates were introduced from other locations within the US, such as from neighboring states.
As minor points:
1 the title seems quite long, usually a short title is much better memorized by readers, may be "High throughput sequencing-aided survey reveals" could be deleted?
Response: We prefer to keep “High throughput sequencing-aided survey reveals" since identification of viruses by HTS is an important part of the article. This will reflect the contents of our article more completely and accurately.
2 in the abstract, what does "n" mean? (= haploid number of chromosomes?) As apparently you do not use this in the text, is it necessary? Moreover, normally you should use this only if you checked by yourself the number of chromosomes (?), and as you experimented with commercial material you cannot be sure at all of the number of chromosomes without checking them by observation of mitosis cells… [believe me, polyploidy is very (extremely…) frequent in agronomy selections]; homogenize with points 4 and 5 just below;
Response: ‘n’ means the number of samples.
3 in the introduction, as “Cucurbitaceae” is already a family taxonomical level (ending with “…ceae”), you cannot use it with “family” together, put” family” somewhere else in the sentence; change this also in the discussion line 274 (“in the family Cucurbitaceae and Fabaceae”);
Response: We do not quiet understand what this means. However, we have made changes and separated family from ‘ceae’
4 in the introduction, put (at least once in the text) the name(s) or abbreviation(s) of the author(s) of latin names of the plants (use international plant names index IPNI internet or another relevant up-to-date site); do the same for the viruses as you use or not details of genus taxonomical rank and it has to be homogeneous; precise also that Bemicia tabaci is an insect; precise also what are “zucchini and beans”, for non-specialist it will be very helpful; homogenize with points 2 and 5 just after;
Response: Latin names have been added for all the plants.
5 related with point 4 just above, in material and methods (or in the introduction) precise if the plants are cultivars, subspecies… it may influence your discussion if your collection is heterogeneous and/or not for one single taxon; cultivars are not wild taxon, they may have different chromosome numbers; homogenize with points 2 and 4 just before
Response: There can certainly be variation in distribution of these viruses on different cultivars. However, this was not the objective of our study and we did not have information on cultivars or varieties. Our study was focused on identifying the prevalence and distribution of viruses on cucurbits in Georgia under field conditions.
6 in the results 3.1, are all these symptoms in the figure 1?
Response: Yes, All the symptoms are included.
7 in the discussion line 295, is “Crinvirus” correct?
Response: It is Crinivirus. We have corrected it.
8 in the discussion, precise if from the paragraph “Most damage due to WTVs is…” it concerns your experiments as it is not enough clear;
Response: It is WTVs. We have defined this on L274.
9 in the discussion line 328, what is WFT virus? For non-specialist it has to be clarified;
Response: It is WTVs. We have defined this on L274.
10 for figure 2, add a horizontal title for the plants and a vertical title for viruses, it will be more rapid to check;
Response: X axis and Y axis title added.
11 for figure 3, the quality of the figures is not good at all, we cannot read the letters; may be it is better to divide the data in two figures in order to enlarge each set of data;
High resolution image is included
12 for figure 4, use colours to enhance your experimented material.
Response: We avoided adding bright colors so as not to distract the reader from the key message of the figure.
Round 2
Reviewer 2 Report
Dear colleagues,
Thank you for this new version that I was very glad to read, greatly improved and making this ms almost acceptable concerning my remarks, however still very few minor points have to be considered:
- For my –major-point 1, the response (“ It is difficult to differentiate…) should appear in the 1 Symptomatology part, more or less shortened if needed, however it is important for readers to know this for the present study, as in other virus papers “one virus-one symptom” photos are provided;
- For my major-point 3, the response from “for these viruses, there is a high degree…”, very clearly explained, should be included in the discussion part, more or less summarized if needed, as it represents a very fruitful result in itself;
- In the minor points, “n” should appear linked with “total number of samples (n)” in table 3 and supplementary table 2, it will make these tables much more readable;
- Latin names of plants appear now, but still apparently not the name or abbreviations of the authors, this is very important for taxonomy as taxa may have various synonyms.
Author Response
Thank you for the constructive and critical comments. We accepted the suggestion and modified the manuscripts accordingly.
- For my –major-point 1, the response (“ It is difficult to differentiate…) should appear in the 1 Symptomatology part, more or less shortened if needed, however it is important for readers to know this for the present study, as in other virus papers “one virus-one symptom” photos are provided;
Modification: Included on line 206-208 “The viruses that were identified in this study and mentioned in the following sections exhibit overlapping symptoms and are difficult to differentiate in mixed infection under natural conditions.”
- For my major-point 3, the response from “for these viruses, there is a high degree…”, very clearly explained, should be included in the discussion part, more or less summarized if needed, as it represents a very fruitful result in itself;
Included in 414-417 “]. The isolates from Georgia are closely related to isolates from many areas of the world, which make it difficult to determine the original source of introduction other than most likely these were introduced from other locations within the US, such as from neighboring states.”
- In the minor points, “n” should appear linked with “total number of samples (n)” in table 3 and supplementary table 2, it will make these tables much more readable;
The sentence in abstract is modified “Symptomatic samples were collected and small RNA libraries were prepared and sequenced from three cantaloupes, four cucumbers and two yellow squash samples”
- Latin names of plants appear now, but still apparently not the name or abbreviations of the authors, this is very important for taxonomy as taxa may have various synonyms.
Latin names for the plants have been updated.
cucurbits (Cucurbitaceae Juss.)
cantaloupe (Cucumis melo var. cantalupensis Naudin)
cucumber Cucumis sativus L.
yellow squash and zucchini Cucurbita pepo L.
beans Phaseolus vulgaris L.